

# Local spatial variability in the occurrence of summer precipitation in the Sør Rondane Mountains, Antarctica

Alfonso Ferrone[1,2], Étienne Vignon[3], Andrea Zonato[4], and Alexis Berne[1]

[1]Environmental Remote Sensing Laboratory, École Polytechnique Fédérale de Lausanne (EPFL), Lausanne, Switzerland
[2]MeteoSwiss, via ai Monti 146, Locarno, Switzerland
[3]Laboratoire de Météorologie Dynamique/IPSL/Sorbonne Université/CNRS, 8539 UMR, Paris, France
[4]Royal Netherlands Meteorological Institute (KNMI), De Bilt, The Netherlands

**Correspondence:** Alexis Berne (alexis.berne@epfl.ch)

**Abstract.** During the austral summer 2019/2020, three vertically-pointing K-band Doppler profilers (MRR-PRO) have been deployed along a transect across the Sør Rondane Mountains, directly south of the scientific base Princess Elisabeth Antarctica. The MRR-PRO have been placed at locations corresponding to different stages of the interaction between the complex terrain and the typical flow associated with precipitating systems. The radar measurements, alongside information from the ERA5 reanalysis and a set of high-resolution WRF simulations, have been used to study the spatial variability of snowfall across the transect. Radar observations reveal differences in the frequency of occurrence of virga and surface precipitation above the transect. An analysis of the WRF outputs reveals the presence of a relatively dry layer above the radar locations, reaching a constant altitude of 3.5 km above mean sea level. Due to the complex terrain, the depth of the layer varies across the transect, affecting sublimation and the occurrence of virgae. Combined information from the ERA5 reanalysis, the WRF simulations, and ground-level measurements suggest that orographic lifting enhances precipitation above the highest mountain peaks. Finally, the analysis of the succession of virga and surface precipitation above the sites shows that, in most cases, they represent different stages of the same large-scale events. This study reveals the significant spatial variability in the occurrence of precipitation in a region of complex terrain, emphasizing the importance of collecting snowfall measurements in the mountainous regions of the Antarctic continent.

## 1 Introduction

The Antarctic continent is almost completely covered by an ice sheet that, with its estimated volume of 27 million $km^3$, contains about 70% of Earth's freshwater (Fretwell et al., 2013). The topic of its mass balance has been included among the priorities in the study of the continent (Kennicutt et al., 2014), given the positive contribution of the ice loss to the sea level rise, which may increase in the future (Ritz et al., 2015; DeConto and Pollard, 2016; Noble et al., 2020). Snow precipitation is the largest positive contributor to the mass balance (Van Wessem et al., 2018). Nonetheless, it is a complex physical process, particularly difficult to monitor in a remote and harsh environment such as Antarctica. Its estimates can be obtained from a wide set of sources, ranging from numerical models (Bromwich et al., 2011), satellite-based measurements (Palerme et al., 2014) or in-situ scientific instruments, recording information at the surface (König-Langlo et al., 1998) or in a wider volume through



remote sensing (Konishi et al., 1998). Depending on the location and scale of the phenomena being investigated, each of these
methods has a varying degree of usefulness and a certain set of limitations. For instance, the higher viewpoint of satellites allows
them to collect measurements repeatedly over large regions of the continents, which would otherwise be almost inaccessible,
making them particularly suitable for continent-wide studies (Palerme et al., 2014). However, this downward-looking geometry
constitutes an obstacle when the measuring instrument is a meteorological radar, due to the blind range near the ground (Maahn
et al., 2014), which limits the capability of sampling the precipitation over the full atmospheric column (Palerme et al., 2017).
Ground-based radar installations, on the other hand, offer a privileged viewpoint on the lower parts of the atmosphere, being
instead limited in the spatial coverage of their measurements. The complexity and cost associated with the deployment and
maintenance of these instruments in polar conditions resulted in a low number of such installations on the Antarctic continent,
especially inland and far from scientific bases.

For these reasons, the availability of multiple datasets of ground-based instruments collected in the complex terrain sur-
rounding the Belgian base Princess Elisabeth Antarctica (PEA) during the PEA Orographic Precipitation Experiment (POPE)
is a unique opportunity to study precipitation over the region, especially in proximity to the ground, where alternative data
source would be unavailable. The POPE campaign took place between November 2019 and February 2020, and the informa-
tion on the instruments deployed and the datasets collected is presented in Ferrone and Berne (2023a). The main novelty of
this campaign is the deployment of multiple profilers in remote sites across the region of the Sør Rondane Mountains, directly
south of PEA.

The base is located in the Queen Maud Land, a sector of East Antarctica bounded between $20°$ W and $45°$ W of longitude.
This area has been the subject of multiple studies highlighting the major role of few considerable precipitation events in the
total accumulation (Braaten, 2000; Schlosser et al., 2010; Fujita et al., 2011) and the consequent effect on the surface mass
balance (Boening et al., 2012). In this region, the snow accumulation has been increasing in the past century, mitigating the
contribution of the ice sheet to sea level rise (Medley et al., 2018; Turner et al., 2019).

POPE is not the first measurement campaign to deploy weather radars in the region. It has been preceded by the installation
of a Micro Rain Radar (MRR-2), a vertically-pointing K-band (24 GHz) Doppler weather radar (Klugmann et al., 1996), on
the rooftop of PEA as part of the HYDRANT project (Gorodetskaya et al., 2015). The profiles collected by this instrument
have been central in studying precipitation above the base (Souverijns et al., 2017, 2018). In particular, these measurements
evidenced the frequent occurrence of virga (i.e. precipitation that sublimates before reaching the ground) over PEA (Durán-
Alarcón et al., 2019). The low-level sublimation of precipitation has already been the subject of studies in different locations
on the Antarctic continent (Grazioli et al., 2017), demonstrating the usefulness of radar measurements for characterizing the
phenomenon.

In this study, we investigate the interaction between precipitation and the mountainous terrain immediately south of PEA,
where measurements were previously unavailable. In particular, we investigate the spatial variability in the occurrence of virga
and surface precipitation thanks to a transect of K-band Doppler profilers, deployed in three sites across the Sør Rondane
Mountains, up to a distance of 30 km from PEA. High-resolution numerical simulations performed with the Weather Research
and Forecasting (WRF) model (Skamarock et al., 2021) help us understand the complex interactions between precipitation



systems, local orography, and boundary-layer flows. The ERA5 reanalysis (Hersbach et al., 2020) also allows us to characterize
the synoptic conditions that preferentially lead to snowfall and/or virga over the three remote sites.

The study is structured as follows. Section 2 illustrates the data used in the study, and the methods employed in their
processing. Section 3 presents the results and discusses them. Section 4 concludes the study.

## 2 Data and methods

### 2.1 MRR-PRO

The measurements collected by three MRR-PRO deployed along a transect across the Sør Rondane Mountains, reaching a
maximum distance of 30 km from PEA, are central to the present study. The locations of the three sites and relevant topographic
features are shown in Figure 1. The three profilers and their deployment sites will be denoted by the serial number of each
radar, similarly to the article that presents the POPE campaign (Ferrone and Berne, 2023a). While the latter comprehensively
presents the instruments and the datasets, here we will summarize the information relevant to the current analysis:

- the MRR-PRO 23 has been installed at 1543 m above mean sea level (a.m.s.l.), the lowest altitude in the transect, in a
  valley oriented along the North-South axis, in which the Gunnestadbreen glacier lies. The site is at the eastern edge of a
  mountain complex, which can be considered upstream of it in the typical north-easterly flow associated with significant
  snowfall at PEA.

- the MRR-PRO 06 was placed in a high valley directly North of the mountain Widerøefjellet, at 2000 m a.m.s.l., approx-
  imately 6.5 km from the previous site.

- the MRR-PRO 22 has been deployed at the onset of the Antarctic plateau, at 2360 m a.m.s.l., south (and therefore
  downstream) of the mountains. It is the highest of the three locations, and its distance from the MRR-PRO 23 and 06 is
  respectively equal to 17 km and 13 km.

Alongside each MRR-PRO, an automated weather station (Vaisala model WXT536 for the MRR-PRO 22 and MRR-PRO 23,
WXT520 for the MRR-PRO 06) has been installed, and its measurements of wind direction are used in the present study. A
pyranometer and a pyrgeometer (Kipp & Zonen, model CMP3 and CGR3 respectively) accompany the MRR-PRO 06 and
provide information on the downwelling radiative fluxes in the visible and infrared spectrum.

The duration of all datasets has been restricted to the period in which all three instruments were deployed and collecting
measurements: from 15 December 2019, at 00:00 UTC, to 17 January 2020, at 12:00 UTC. Given the limited period covered
by the measurements, the precipitation accumulation and frequency of occurrence recorded during this time interval have been
compared to the same period in the previous 20 austral summers. The analysis is presented in Appendix A.

The radar data have been processed and postprocessed using the "Enhancement and Reconstruction of the spectrUm for the
MRR-PRO" (ERUO hereafter) algorithm (Ferrone et al., 2022). This re-processing aims to remove non-meteorological artifacts
from the measurements and to improve the sensitivity of the radar variables. Despite this re-processing, further steps are





required for ensuring the quality of the derived data products. The following subsections describe the additional postprocessing, the time-averaging of the radar variables, and the detection of virga and surface precipitation above the three sites.

### 2.1.1   Limits on the vertical extent of the profiles

The first of the radar variables central to this analysis is the equivalent attenuated reflectivity factor, $Z_{ea}$. Due to measurement inconsistencies in the first 300 m above the instruments, we follow Ferrone et al. (2022) and exclude the first 300 m from the

analysis of radar profiles. A similar value for the lowest reliable range gates has been used in previous studies using the MRR-2 (Durán-Alarcón et al., 2019).

    For the MRR-2, this limit was dictated by issues in the first couple of range gates, each covering 100 m of the profile. In our MRR-PRO datasets, the behavior may be caused by near-field effects, or the typical drop in power visible in the raw Doppler spectra at the lowest range gates, an intrinsic characteristic of the measurements from this radar.

Another limitation of this MRR-PRO dataset is the existence of artifacts in the measurements. A particular type of these non-meteorological returns is interference lines, characterized by significant peaks in the raw Doppler spectra at a fixed height above the radar, visible in both clear-sky and precipitation conditions. Given their near-constant presence, even in absence of external obstacles, we hypothesize that their origin is internal to the radar. Their characteristics are further discussed in Ferrone et al. (2022).

An interference line, not completely removed by the ERUO algorithm, is visible in the MRR-PRO 06 dataset, at approximately 4.5 km above the first range gate. Since Ferrone et al. (2022) shows that none of the three radars recorded meteorological signals above the first 3 km of the profile, a maximum height limit of 3 km above the first range gate of each MRR-PRO has been imposed on the three datasets. The interference line of the MRR-PRO 06 appears above this height limit, and it is therefore excluded from the rest of the analysis.

### 2.1.2   Reflectivity offset correction

A comparison of the three MRR-PRO datasets reveals the existence of a significant difference between the $Z_{ea}$ values collected by the MRR-PRO 22 and those from the other two radars in the transect. The likely cause for this difference is a miscalibration of the MRR-PRO 22. Due to the time constraints of the field campaign, we could not directly test the agreement between the three MRR-PRO before or after the campaign at PEA. This lack of comparable measurements limits our ability to precisely

correct this significant offset between the radars.

    Therefore, we decided to follow a different approach, estimating the offset from the sensitivity curve of the three MRR-PRO. By following this method, we assume that the sensitivity of the three radars should be approximately the same across the profiles since their electronic is the same. Any systematic deviation between the profiles becomes our estimate of the offset.

    Following Ferrone and Berne (2023a), an estimate of 10 dB has been used for the value of the offset, which has been added

to the $Z_{ea}$ values collected by the MRR-PRO 22. The main reason why this offset has been computed is to allow the usage of the same threshold on $Z_{ea}$ for postprocessing the three datasets. The uncertainty on the proposed value was deemed too large for comparing derived quantities, such as estimates of precipitation rates, in the context of the current study. Therefore





the analysis presented in the following sections focuses on the occurrence of precipitation at each location, rather than on the accumulation or snowfall rate.

### 2.1.3 Removal of leftover non-meteorological artifacts

Despite the ERUO postprocessing, a few leftover non-meteorological artifacts still affect some of the radar measurements, especially in the MRR-PRO 06 dataset. It should be noted that the amount of interference lines and spurious signals in the raw spectra varies between different MRR-PRO systems. The ERUO postprocessing does not take an aggressive approach to noise removal, prioritizing the preservation of any sensitivity improvement. A cleaner set of variables can be achieved at the cost of losing some of the faintest or most isolated meteorological signals.

The current analysis, however, requires a number of false precipitation signals as low as possible, to avoid artificial increases in the detection of virga or surface precipitation over the three sites. Therefore, the ERUO products underwent a second noise-removal stage, with stricter conditions than the default ERUO ones. This further refinement starts by imposing two thresholds: the first on the signal-to-noise ratio ($SNR$), $SNR > -10$ dB, and the second on the reflectivity factor, $Z_{ea} > -10$ dBZ.

The particularly low value of the minimum accepted $SNR$ value is dictated by how this variable is computed in our dataset. $SNR$ is derived from the ratio between the integral of the signal and the noise floor. The latter is usually high in the MRR-PRO data and, while the peaks associated with the meteorological signal are often clearly distinguishable, their prominence above the noise level can be smaller than the noise level itself. In general, the spurious peaks associated with the leftover non-meteorological signal in the measurements are characterized by low values in both $SNR$ and $Z_{ea}$, often at the limits of the detection capabilities.

To choose the exact value of the two thresholds, we used the information provided by the pyranometer co-located with the MRR-PRO 06. This instrument measured the downwelling solar irradiance, giving us a way to distinguish between clear-sky and cloudy conditions above the site. Selecting non-null measurements from the MRR-PRO 06 dataset in clear-sky conditions, we were able to identify the typical range of values for the leftover noise in the postprocessed radar data. The chosen thresholds exclude approximately 91% of this noise. From a visual inspection of their time series, it appears that the chosen threshold removes most of the noise also in cloudy conditions. The MRR-PRO 22 and MRR-PRO 23 datasets are less affected, and the amount of spurious signals visible in these datasets is lower than in the MRR-PRO 06 case after the thresholds are applied to both of them.

### 2.1.4 Temporal averaging

The temporal resolution of the MRR-PRO measurements is equal to 30 s. This value has been chosen as a compromise between the importance of monitoring relatively fast changes in precipitation and the need for a long integration time to improve the sensitivity.

At this resolution, large-scale dynamics and turbulence can significantly contribute to the variability in the observed radar variables. Therefore, we decided to average them over longer periods, using a sequence of window sizes starting at 15 min and ending with 60 min, with increments of 15 min. The averaging of $Z_{ea}$ has been performed in linear units and then converted



back to logarithmic form. The procedure is analogous to the one described by Welsh et al. (2016), and later applied to the study of MRR-2 profiles by Durán-Alarcón et al. (2019). Given the limited duration of our datasets, window sizes above 60 min have been tested but not used in the subsequent analysis due to the low number of time steps remaining after the averaging.

An inspection of the resulting time series reveals the existence of a few instances in which non-meteorological noise is still visible in the averaged variables, despite the removal described in the previous subsection. If enough noise artifacts appear at the same range gate in the span of a single temporal window, the averaged noise will appear in the final variables. Fortunately, it is an unlikely occurrence, and noise in the final variables appears as spurious non-null values isolated between each other.

    The first step implemented to remove these artifacts is the exclusion of signals with a duration shorter than 5 min, at each range gate, from the averaging. This condition takes advantage of the typical temporal continuity of meteorological signals.

To exploit the continuity in time and space after the averaging, an approach similar to the one described by Ferrone et al. (2022) for the post-processing has been followed. The averaged measurements can be considered analogous to a 2-dimensional matrix, with time on one axis and the vertical distance above the radar on the other. In this matrix, the number of contiguous, non-null measurements can be counted using the image processing capabilities of the Scipy python library (Virtanen et al., 2020). We imposed a minimum threshold equal to 3 to this number, excluding in this way isolated artifacts that show no continuity with significant precipitation systems.

    This procedure may remove some measurements immediately before or after precipitation, which may lack temporal and spatial continuity with the core of the meteorological signal. To avoid this undesirable behavior, the removal is performed only if fewer than 5 valid measurements are present at the current time step or in the ones immediately before or after.

### 2.1.5   Detection of virga and surface precipitation

For each of the four chosen averaging windows, all the time steps containing at least one valid measurement have been divided into two categories: virga and surface precipitation. To distinguish between the two, we checked whether the signal was detected at the lowest valid range gate, 300 m above the radar. If the valid measurements in the profile reach this lowest range gate, the time step is labeled as surface precipitation, otherwise as virga. The approach is analogous to the one presented in Durán-Alarcón et al. (2019) for the MRR-2.

It is theoretically possible that some of the measurements labeled as surface precipitation do not reach the true ground level, becoming virga in the missing 300 m of the profiles. Therefore, it should be noted that surface precipitation corresponds in our case to the 300 m precipitation. However, the inclusion of these last meters may cause two major problems. Firstly, as described by Ferrone et al. (2022), low-reflectivity measurements are less likely to be recorded in this lower region. Therefore, faint precipitation may be undetectable close to the ground, artificially increasing the number of virga events detected. Additionally,

the sensitivity change in this region is not the same for the three MRR-PRO. This behavior could introduce artificial differences in the statistics computed for three MRR-PRO, compromising the validity of the analysis. With the 300 m height limit in place, the comparison becomes fairer due to the more consistent behavior in the remaining vertical extent.



## 2.2 Numerical simulations with the WRF model

We also used numerical simulations carried out with version 4.1.1 of the WRF model. The simulation configuration has been developed for precipitation studies in Antarctica (Vignon et al., 2019a) and can be summarized as follows. WRF is run with a downscaling approach with a parent domain of 27 km resolution containing three (one-way) nested domains with 9, 3, and 1 km resolution centered over PEA (see Figure 2). The topography is from the Bedmap2 1-km resolution dataset from Fretwell et al. (2013). The lateral forcings, the sea ice cover, the sea surface temperature, and the initial conditions are from the ERA5 reanalysis (Hersbach et al., 2020). To ensure realistic large-scale dynamics throughout the simulation period, the 27-km resolution domain has been nudged above the boundary layer towards ERA5 reanalysis for zonal and meridional wind with a relaxation time scale of 6 h. The cloud parameterization used is the Morrison 2-moment microphysical scheme (Morrison et al., 2005) modified with a new ice nucleation parameterization adapted to the low concentrations of ice nuclei particles in the Antarctic atmosphere (Vignon et al., 2021). The other physics options include the new version of the Rapid Radiative Transfer Model for General Circulation Models (Iacono et al., 2008) radiation scheme for longwave and shortwave spectra, the Noah land surface model (Niu et al., 2011) with adaptations by Hines and Bromwich (2008) and the Mellor-Yamada-Nakanishi-Niino planetary boundary layer scheme (Janjić, 1994) coupled with its associated surface layer scheme. The Kain-Fritsch cumulus scheme (Kain and Fritsch, 1990) is activated for the 27-km resolution domain. Otherwise mentioned, the model variables shown in this paper are from the 1-km resolution domain.

The first of the variables taken into consideration is the horizontal wind speed. Three sets of profiles have been extracted from the model grid points closest to the three MRR-PRO locations in the transect. The median and interquartile range (IQR) of the variable have been computed from the time series of profiles, covering the same temporal extent as the three MRR-PRO datasets.

The second variable extracted from the WRF simulations is the relative humidity with respect to ice ($RH_i$), computed in post-processing from the specific water vapor content and the temperature of the model using the formula for the saturation vapor pressure with respect to ice from Huang (2018). This variable allows us to link eventual layers of relatively dry air to the virga observed in the radar measurements. As for the wind speed, profiles have been extracted above the three transect sites and the median and IQR have been computed for the period in which the MRR-PRO data are available.

In addition to these profiles, the total accumulation of solid precipitation at the ground has been used in the analysis. The mixing ratio of graupel and snow, together with the vertical wind speed simulated by the model, has also been used to estimate the precipitation flux at a fixed altitude.

## 2.3 ERA5 reanalysis

The current study uses the wind direction at 500 hPa and the sea level pressure from the ERA5 reanalysis, on its native $0.25° \times 0.25°$ horizontal grid, at an hourly temporal resolution (Hersbach et al., 2020). The wind direction has been extracted at the grid point closer to PEA.



To investigate whether a link exists between the advection of the cyclonic systems alongside the coastline and the presence of virga and surface precipitation systems over the MRR-PRO transect, we identify low-pressure systems transiting over the Southern Ocean from the ERA5 reanalyses. The detection is performed by the algorithm developed by Jenkner et al. (2010). The same technique has been already been used by Sprenger et al. (2017) on ERA5 data, and it is the same one adopted by Jullien et al. (2020) in their analysis of the atmospheric dynamics during precipitation events in Adélie Land.

For each MRR-PRO, the virga and surface precipitation labels determined in section 2.1.5 are assigned to these time steps. The virga category can be further refined, by separating the virgae that precede surface precipitation from those that follow it. The distinction between the two is performed by identifying, for each virga occurrence, the time-step containing surface precipitation that is closer to it in time. The label is assigned by checking whether this time step is before or after the virga. In case the time difference between the virga and the closest surface precipitation before and after it is the same, no label is

assigned to the virga.

## 3    Results and discussion

### 3.1    Occurrence of virga and surface precipitation

The labeling of virga and surface precipitation time steps presented in section 2.1.5 permits the computation of simple statistics related to the occurrence of the two weather states above the three instrumented sites along the transect. Figure 3 shows the

results for all four temporal windows used for computation, denoted by vertical bars of different colors.

Panel 3.a shows the ratio between virga and surface precipitation at the three locations in the dataset. MRR-PRO 23 stands out for the relatively high ratio of virgae to surface precipitation (from 65 to 85%, compared to typically 40 and 25% for the MRR-PRO 06 and MRR-PRO 22), with virgae constituting between 40% and 50% of all the valid precipitation signals, depending on the window size, as illustrated by panel 3.c. Interestingly, panel 3.e demonstrates that the absolute number of

virga time steps for MRR-PRO 23 is not significantly higher than what we observe for the other two radars. Instead, as visible in panel 3.f, the lack of surface precipitation is likely the main factor explaining the high ratios in panels 3.a and 3.c.

The ratios of virga to surface precipitation for the MRR-PRO 06 and MRR-PRO 22 are similar but it is worth noting that MRR-PRO 06 records an overall higher number of time steps containing valid meteorological signals, as shown by panel 3.b. A closer look at panels 3.e and 3.f reveals that the MRR-PRO 06 experiences virga for a substantially higher number of time

steps than the MRR-PRO 22, and surface precipitation for a slightly higher one, especially for averaging windows longer than 15 min. Combining information for the three sites, we can conclude that the ratio of virga to surface precipitation in the transect decreases with the altitude of the site.

Many of the ratios shown in Figure 3 depend significantly on the averaging window. This dependence is visible for all three MRR-PRO, even though it varies between systems. Generally, a wider averaging window increases the number of time

steps containing valid signals. This can be caused by the discontinuity in time of the precipitation measurements. Valid signal spanning only a fraction of a large averaging window will still result in an averaged time step containing precipitation. A smaller set of windows may cover the same interval, with only some of them containing enough valid measurements to result




in a non-null averaging result. Despite this variability, the order in which the three MRR-PRO can be ranked based on the ratios displayed in Figure 3 seldom changes with the size of the averaging window.

## 3.2 Vertical structure of precipitation from radar measurements

We now gain better insights into the spatial variability of surface precipitation and virga by examining the vertical structure thereof from a careful inspection of the empirical distribution of radar variables. For the subsequent analysis, a 30 min temporal window has been chosen, as a trade-off between a sufficiently long window that averages enough signal to compute robust statistics on the one hand and sufficiently short windows to capture the time variability of precipitation during weather events on the other hand. Note that additional analyses with other window sizes revealed that our results do not depend significantly on the choice of the averaging interval (not shown).

### 3.2.1 Equivalent attenuated reflectivity factor

The first radar variable to be examined is the equivalent attenuated reflectivity factor. The distribution of all $Z_{ea}$ profiles resulting from the chosen averaging windows can be seen in the 2-dimensional histograms, in the first row of panels of Figure 4. The histograms are strongly affected by the limited sensitivity of the instruments at higher altitudes, as demonstrated by the proximity of the sensitivity curve (dotted green line) to the lower limit of the empirical distributions. The lack of lower reflectivity values in those range gates skews the distribution, resulting in a median value that increases with height. This behavior limits our ability to interpret the upper part of the profiles in terms of microphysical processes.

The panels in the bottom row show the median and interquartile range when virga and surface precipitation are distinguished. It is immediately possible to spot an interesting behavior in the lower section. At approximately 3.5 km a.m.s.l. (purple scale on the right of each plot) the two sets of curves start to diverge, with the virga event consistently recording increasingly lower reflectivity values. Given the difference in altitude between the three MRR-PRO, the separation occurs at a higher range gate number for the MRR-PRO 23, corresponding to approximately 2 km above the instrument. This marked drop in $Z_{ea}$ is likely due to the occurrence of sublimation in this lower section, as can be expected during virga events. Similar features have contributed to the identification of sublimation in previous scientific studies at both PEA (Durán-Alarcón et al., 2019) and Dumont d'Urville coastal station (Grazioli et al., 2017).

### 3.2.2 Spectral width

It is worth mentioning that the analysis of the profiles of Doppler velocity ($V$) reveals no significant difference between virga and surface precipitation (not shown). Nonetheless, the profiles of time-averaged spectral width ($SW$) differ near the ground between virga and surface precipitation cases. At approximately 3.5 km a.m.s.l. the median values of virga and surface precipitation events start to diverge, even though the separation of the two lines is arguably less noticeable than for the reflectivity (Figure 4). Interestingly, the surface precipitation events are characterized by higher $SW$ values for most of the section below





3.5 km. This behavior may indicate a more turbulent environment or a broader particle size distribution in this layer during precipitation.

In the lowest layers, an increase in $SW$ with decreasing height can be seen for virga events for MRR-PRO-06 and MRR-PRO-22, whose values reach or sometimes overcome those recorded during surface precipitation. We suspect that turbulence may play a more significant role in this lower section of the profiles. A possible explanation for this behavior could be the presence of a larger speed and directional wind shear during virga cases, even though we could not verify this interpretation through measurements or numerical simulations.

Note that for both $Z_{ea}$ and $SW$ profiles in the MRR-PRO 23 dataset, two sudden increases can be observed between 1.0 km and 1.5 km. Two interference lines were present in the radar measurements at those heights before the post-processing. These increases are likely the leftover effects of those lines, caused by an imperfect reconstruction of the spectra by ERUO, resulting in a slight overestimation of both variables.

### 3.3    Time elapsed between virgae and surface precipitation

Jullien et al. (2020) show that virga and surface precipitation at Dumont d'Urville station are associated with the same precipitating system but they correspond to different phases of the event. It is worth investigating if this is also the case along our transect in the Sor Rondane Mountains.

The temporal relationship between virgae and surface precipitation can be quantified by computing the time elapsed between each virga time step and the closest occurrence of surface precipitation. The histogram summarizing these time differences for
each of the MRR-PRO sites is displayed in Figure 6. The position of the quantiles 0.1 and 0.9 in all the panels shows that the two precipitation types take place during the same day in the large majority of the events, similarly to the situation described by Jullien et al. (2020) for the coastal location of Dumont d'Urville. However, the near-zero median difference in the top row of panels suggests that virga both precedes and follows precipitation at the ground during a typical precipitation event at the three sites. The asymmetry in the position of the quantiles with respect to the 0 h difference, visible in the panels 6.a- 6.c, may
reflect differences in the continuity of the meteorological signal for virga before and after the surface precipitation event.

The correlation between the occurrence of meteorological signals (virga or surface precipitation) at each couple of sites along the transect has also been investigated. The resulting Pearson correlation coefficient varies between 0.6, for the couples MRR-PRO 06/22 and MRR-PRO 06/23, and 0.5 for the MRR-PRO 22/23.

To account for the propagation of weather systems, the correlation has been computed again after adding a series of time
lags, varying between -2 hours to + 2 hours, to the time of occurrence of the meteorological events at each MRR-PRO site. Since the detection of virga and surface precipitation has been performed on averaged $Z_{ea}$ values, the time lags follow the same temporal discretization as the averaging windows. Among the latter, only the 15 min one is short enough to capture the delay in the onset of precipitation between some of the sites, given the proximity of the radars.

For the couples MRR-PRO 06/22 and MRR-PRO 22/23, the maximum Pearson correlation coefficient is obtained when
the MRR-PRO 22 data are delayed by 15 minutes. A similar behavior can be observed for the MRR-PRO 06/23, for which the maximum correlation is reached when the measurements of the MRR-PRO 06 are delayed by 15 minutes. However, the





increases compared to the 0 min time lag case are almost negligible, with values between 0.001 and 0.003. Larger values of the time lag result in a lower Pearson correlation coefficient for all MRR-PRO couples, with the minimum always reached for a time lag of $\pm 2$ hours. Overall, these results suggest a southward propagation of the weather systems.

### 3.4 Investigation of the low-level snowflake sublimation from WRF simulations

We now use the WRF simulations to investigate the conditions responsible for the sublimation identified in the radar measurements. As the intensity of snowflake sublimation strongly depends on wind speed and relative humidity, we first analyze the profiles of horizontal wind speed and relative humidity with respect to ice extracted from the WRF simulations. The profiles of the median and interquartile range are shown in Figure 7. The profiles have been truncated at a maximum height of 5 km

a.m.s.l., given the limited vertical extent of the MRR-PRO data to which the model data are compared with.

Similarly to the distributions of radar variables presented in the previous sections, we decided to divide the profiles depending on the simulated weather states, defined as follows:

- Surface precipitation events (green continuous curves), in which the accumulation at the ground level simulated by WRF has been used to detect the suitable time steps;

- Virga events (orange dashed curves), in which no accumulation is recorded at the lowest level of the model and a positive mixing ratio of snowfall or graupel exists in the profile. The second condition must be verified for at least 5 levels, to ensure consistency with the detection of virga and surface precipitation in the MRR-PRO data;

- Clear-sky conditions (purple dot-dashed curves), in which both the accumulation at the ground and the mixing ratio of snowfall or graupel in the model are equal to zero.

The discrimination between precipitation types is performed by using the simulated profiles and not the radar observations to limit the impact of temporal mismatch between the simulated weather and the real one. Differences between the time at which virga or surface precipitation appears in the model and in the radar observations may result in the inclusion of profiles in the analysis that does not reflect the condition behind the phenomenon that we are investigating. For instance, a delay in the appearance of surface precipitation in the WRF outputs may result in the inclusion of profiles of relative humidity far from the

saturation with respect to ice. A comparison between the simulated occurrence of virga and surface precipitation and the one recorded by the three MRR-PRO is presented in Appendix B.

The statistics for the horizontal wind speed are presented in the first row of Fig. 7. At the three sites, most of the median wind speed profiles experience a maximum between 2.5 km and 3.5 km above sea level. A considerable decrease in wind speed can be observed below this maximum, reaching a local minimum at the ground level.

The similarity between the heights of the local maxima in wind speed reveals the existence, to a certain degree, of uniformity in the circulation in the low troposphere above the three sites of the transect. The high values of horizontal wind speed in the lower part of the profiles, together with the low relative humidity shown in panels 7.d- 7.f, indicates the presence of an outflow from inland, driven by local katabatic effects. The structure of the profiles resembles in shape those observed for coastal




locations by Vignon et al. (2019b), even though the mountain range south of PEA prevented the detection of a katabatic layer

over the base in the same study. The difference between the profiles in panels 7.a-7.c and the ones of the aforementioned study may indicate that significant differences can exist in the structure of the atmosphere over relatively small distances, highlighting the importance of extending the measurement over the mountainous region bordering PEA.

The bottom row of Figure 7 shows the statistics computed from the profiles of relative humidity with respect to ice. A local maximum can be observed at a similar height to the wind speed case, even though its presence is less marked in a few cases

(e.g. clear sky condition in panel 7.e). As can be expected, $RH_i$ values vary significantly across weather conditions, with values close to (or slightly above) saturation visible only during surface precipitation events and at the height of the local maximum for virga events.

A region with lower relative humidity values can be observed in the lowest sections of the $RH_i$ profiles. This region can be interpreted as a sub-saturated or relatively drier layer in case of precipitation, or simply a dry layer in clear-sky conditions.

Since the altitude of the upper boundary of this layer appears almost constant across the transect, it seems that the height of the terrain controls its depth.

This dry layer can be visualized by examining an example extracted from the precipitation event that occurred on January 6, 2020, shown in Figure 8. We extracted the $RH_i$ values above all grid points intersected by the line connecting the three MRR-PRO sites. A padding of 5 km (i.e. 5 grid points) has been added on each side of the line to investigate the variability

in the direction perpendicular to the transect direction. Panel 8.a illustrates the uniformity of the relative humidity field in the lower atmosphere along the transect, highlighting the differences in the depth of the dry layer above the three sites. Similarly, Panel 8.b shows the IQR of $RH_i$ along the perpendicular direction. Despite the complex terrain, as suggested by the difference between the minimum and maximum terrain height, the IQR is almost always below 5% in the lowest section of the atmosphere, indicating that the horizontal variability is small in comparison to the vertical one. Therefore, we don't expect hydrometeors

falling through the dry layer to experience $RH_i$ values significantly different from the ones shown in Figure 7 as their falling motion deviates from a vertical one.

The precipitation fields simulated by WRF can help us understand the potential impact of this dry layer on the falling hydrometeors above the three MRR-PRO sites. Figure 9 depicts the total accumulation at the ground (panel 9.a) and the estimated precipitation flux at 3.5 km a.m.s.l. (panel 9.b), which corresponds to the top of the dry layer. In the estimation of

the precipitation flux, we assumed a constant 1 m/s fallspeed for the solid hydrometeors, to which we added the vertical wind velocity simulated by WRF. While this approach only provides us with an estimate of the flux, the following analysis does not rely on the precise value at each grid point, focusing instead on the patterns observable above the transect and the surrounding mountains.

Concurring with the radar observations, the grid point nearest to the MRR-PRO 23 site receives the lowest amount of pre-

cipitation at the ground. Panel 9.a also shows that the accumulation values at the MRR-PRO 06 and MRR-PRO 22 sites are similar, with the latter receiving a slightly larger amount of precipitation. Overall, WRF simulates local maxima in accumulation above the mountain peaks, with lower values in the valley above the Gunnestadbreen glacier, in the plateau, and in the flat





region neighboring the base. Panel 9.b shows a similar pattern for the flux at the top of the dry layer, even though the maxima cover a smaller number of grid points.

This information allows us to further discuss the link between the $RH_i$ profiles of Figure 7 and the sublimation identified as the likely cause for the drop in $Z_{ea}$ in section 3.2. We hypothesize that the difference in the depth of this layer at the three locations across the transect (because of the varying ground altitude) is driving the variability in the proportions of virga and surface precipitation events. A deeper layer implies a wider vertical extent that precipitation particles need to fall through before reaching the ground. Sublimation is therefore enhanced, in virga conditions, as falling hydrometeors spend more time

in a sub-saturated environment.

     In section 2.1.5, the highest ratio between the number of time steps of virga and surface precipitation is observed for MRR-PRO 23, whose location has the deepest dry layer in the WRF simulations. The ratio for the MRR-PRO 06 and MRR-PRO 22 are considerably lower, with the first experiencing a slightly higher value than the second, reflecting the relative size of the dry layer above the two locations (MRR06 is about 400 m lower in altitude). However, the difference in the precipitation flux,

shown in panel 7.b, indicates that a second mechanism may contribute to the difference in the total number of time steps with a valid signal at the three locations in the transect. The region adjacent to the mountain peaks may experience orographic lifting, which may enhance precipitation at the MRR-PRO 06 and MRR-PRO 22 sites.

### 3.5   Link between large scale circulation and precipitation along the transect

#### 3.5.1   Insights from a wind direction analysis

Figure 10 summarizes the information on the wind direction at the ground, measured by the automated weather station at each MRR-PRO site, and the wind direction at 500 hPa extracted from the ERA5 reanalysis. Each panel displays the frequency of all wind directions during virga and surface precipitation events at the three locations in the transect.

     The dominant wind direction for events in which snowfall reaches the ground appears to be between NNE and ENE, similarly to what was observed in previous scientific studies at PEA (Souverijns et al., 2018). Due to the configuration of the valley in

which MRR-PRO 23 was deployed, the measurements at the ground at this site differ significantly from their ERA5 counterpart. The valley is surrounded by mountains on its eastern and western sides, forcing the flow mostly along the North-South axis.

     The information on the wind direction at 500 hPa can provide some insight into the interaction between the large-scale circulation and precipitation recorded by the three MRR-PRO. In the typical north-easterly flow, the MRR-PRO 23 site precedes the point of interaction between the flow and the mountain range. The MRR-PRO 06, instead, is placed at a higher elevation

point, closer to the mountain peak. At this location, northerly air parcels may experience orographic lifting, explaining the increased precipitation flux shown in 7.b.

     Focusing on the difference in wind direction between virga and surface precipitation events, Figure 10 shows a more pronounced northerly component for the former, more noticeable at the 500 hPa level than at the ground. An increased frequency in the easterly direction, particularly noticeable for the MRR-PRO 22, can also be observed during virga events at the same

level. The more pronounced northerly component during virga events than during precipitation is particularly noticeable at the



MRR-PRO 23 site. Interestingly, the co-located AWS records a significantly higher frequency of southerly wind, coming down from the Plateau, given the orientation of the valley.

### 3.5.2 Spatial statistics of the low pressure system location during precipitation events

In this section, the temporal succession of virga and surface precipitation over the transect is compared to the position of
low-pressure systems alongside the coastline of the continent. Figure 11 displays the typical location of these low-pressure systems during precipitation events, and during the virgae that precede and follow them. Panel 11.a focuses on the longitude of the low-pressure systems, showing the median and interquartile range of their position relative to PEA. Panel 11.b depicts the median latitude and longitude over a map of the region south of PEA.

In the case of the MRR-PRO 22, the virga preceding (following) the surface precipitation is typically associated with a more
western (eastern) location of the low-pressure system. The MRR-PRO 06 shows a similar behavior, even though the median difference for surface precipitation and the virga following it are significantly closer to each other. The relationship between the low-pressure system location and the presence of virga or surface precipitation over the transect may be analogous to the one described for the coastal Antarctic base Dumont d'Urville by Jullien et al. (2020). A similar circulation type along the coastline has also been associated, in Souverijns et al. (2018), with the highest snowfall occurrence at PEA. In our interpretation, as the
low-pressure system moves eastward along the coast, the direction of the flow at PEA changes slightly. Before the cyclone reaches an eastward enough position along the coast to cause a north-easterly flow at the base, its westward location will result in a northerly flow at PEA. Similarly, after the passing of the cyclone, the wind direction at the base will become more easterly. Under this hypothesis, virga would appear shortly before and after surface precipitation in a large enough number of cases, causing the shift in the relative frequency of wind direction observed in Figure 10.

The location of the low-pressure system is slightly different for the MRR-PRO 23. This difference may be linked to the predominance of virga over the valley. Events leading to surface precipitation over the two sites can, instead, be registered as virga by the MRR-PRO 23, due to the deeper sub-saturated layer above the site that enhances sublimation. Therefore, some of the positions of the low-pressure system that are associated with surface precipitation over the MRR-PRO 06 and MRR-PRO 22 may be classified as virga over the MRR-PRO 23.

We can provide a possible explanation of why virgae preceding the surface precipitation are those affected. Before the typical north-easterly flow brings precipitation over the area, dry wind from the plateau is still dominating the low-level circulation in the valley in which the profiler is deployed. As the overall flow stabilizes, becoming north-easterly, the direction recorded in the valley also changes under the influence of large-scale circulation. However, even though the change occurs often enough to cause a significant shift in the distribution, southerly winds are still measured during some of the surface precipitation events
at the MRR-PRO 23 site.

Nonetheless, it is worth noting a limitation of the method employed for the identification of low-pressure systems. Even though the area chosen for the detection covers only a limited region of the southern ocean directly north of PEA, it is possible that a fraction of the cyclones included in the analysis is not directly responsible for the precipitation over the base and its surroundings. Attempts have been made to replicate the analysis over a smaller region, but this causes the detection of the



low-pressure system to fail in multiple instances. The inclusion of these unrelated cyclones may be the reason behind the
large variability in the quantiles 0.25 and 0.75 in panel 11.a. However, a median location of the cyclone directly south of
PEA is consistent with previous scientific literature (Souverijns et al., 2018). This suggests that the unrelated cyclones have a
significant effect only on the lowest and highest quantiles, while the center of the distribution captures the relationship between
the synoptic conditions responsible for the precipitation over the transect and the precipitation type recorded by the radars.

## 4  Summary and conclusions

In this study, we investigate the differences in the precipitation measured by three K-band Doppler vertically-pointing weather
radars across the mountain range south of the Princess Elisabeth Antarctica research base. The three profilers have been
deployed in a transect, at different stages of the interaction between the typical trajectory of precipitation systems and the
orography. Additional information for the analysis is provided by numerical simulations performed using the WRF model and
from the ERA5 reanalysis. The main results of the study are summarized by the conceptual model displayed in Figure 12.

The occurrence of virga and surface precipitation over the three sites in the transect have been investigated using the mea-
surements from the MRR-PRO, averaged over four different temporal intervals. The main result of this first analysis is the
significantly higher number of virga recorded at the MRR-PRO 23 site, located in a valley at the lowest altitude among the
three radars. This large proportion of virga is not caused by a higher absolute number of time steps of virga measurements,
but by a lower occurrence of surface precipitation at the site. By contrast, the MRR-PRO 22, at the highest location in the
transect, records the lowest proportion of virga. The highest absolute number of surface precipitation events is recorded over
the MRR-PRO 06, close to the mountain peaks and west of the ridge that delimits the valley in which the MRR-PRO 23 is
located.

The 2-dimensional histogram of the 30-minutes averaged $Z_{ea}$ reveals a sharp decrease in reflectivity below 3.5 km a.m.s.l.
consistently over the transect. This behavior suggests that sublimation takes place in this lower layer of the atmosphere, af-
fecting different sections of the profiles depending on the height of the site. A similar analysis, repeated for the spectral width,
shows also a difference in behavior between virga and surface precipitation events starting below 3.5 km a.m.s.l. for all sites.,
even though the separation between the two is not as marked as for the $Z_{ea}$ case.

Profiles of relative humidity with respect to ice ($RH_i$) from WRF simulations covering the same period as the MRR-PRO
deployment help explain the radar observations. In both virga and surface precipitation conditions, the simulations reveal the
existence of a sub-saturated or dry layer over all three sites, with a top height consistently between 3 and 3.5 km above the
mean sea level. The layer is represented by the semi-transparent red plane in the scheme of Figure 12. Profiles of horizontal
wind direction confirm the uniformity of the lower atmosphere over the transect region. The low relative humidity and high
wind speed over the region are likely the result of an outflow of air from the plateau, driven by local katabatic effects.

Overall, the combined information obtained from the radar measurements and WRF simulations allows us to propose an
interpretation of the impact of the structure of the lower atmosphere on the sublimation and on the spatial distribution of pre-
cipitation over the region. The lower troposphere over the analyzed section of the Sør Rondane Mountains is relatively uniform



and, in the lowest 3.5 km, often sub-saturated. This lack of saturation results in the sublimation of the falling hydrometeors, eventually causing the virga observed by the three radars. Since the top height of the sub-saturated layer is constant over the region, the terrain controls the depth of this layer. A deeper layer will result in a longer vertical distance that the hydrometeors have to travel in a sub-saturated environment, leading to enhanced sublimation and a higher percentage of virga in the time series recorded by the radar. This mechanism is consistent with the highest fraction of virga over the MRR-PRO 23, the lowest of the sites, and the lowest fraction over the MRR-PRO 22, the highest point in the transect. This different depth, and the resulting longer fall distance for hydrometeors, are shown by the solid black lines and schematized ice crystals above the MRR-PRO in the conceptual model of Figure 12.

The WRF model also simulates local maxima in the precipitation flux at 3.5 km of altitude and in the ground-level accumulation above the mountain peaks. The wind direction at 500 hPa in the ERA5 reanalysis indicates that a north-easterly flow is typically associated with surface precipitation events over the transect. Given the configuration of the mountain range, the interaction between the flow and the steep terrain likely causes orographic lifting, enhancing the precipitation above the highest sites in the transect. This hypothesis is consistent with the local maxima seen by WRF, while explaining the higher amount of valid precipitation signals recorded by the MRR-PRO 06.

Focusing on the difference in wind direction between virga and surface precipitation events, we notice that increased frequencies for the purely northerly and (to a lesser extent) easterly direction can be noticed during virga events. This behavior, together with the short time intervals that separate virgae and surface precipitation, indicate that the two are associated with different stages of the same large-scale precipitation systems.

A low-pressure system over the Southern Ocean, north of PEA, is often a necessary condition for significant snowfall at the base (Gorodetskaya et al., 2013). An analysis of the locations of these cyclones covering the period of deployment of the three MRR-PRO reveals that virgae are more likely to occur at the beginning and end of large-scale precipitation over the transect. The low-pressure system associated with surface precipitation at the three sites is typically located directly south of PEA. Its position location is shifted westward and eastward for virgae that precede and follow the event, respectively. This interpretation is consistent with the variation in wind direction at the ground (measured) and at 500 hPa (from the ERA5 reanalysis) between virga and surface precipitation events. Figure 12 provides an idealized representation of the movement of the low-pressure system (panel in the top right corner) and the corresponding wind direction over the transect.

The analysis also reveals a particularity of the MRR-PRO 23 site. The distinction between the locations of the low-pressure systems associated with surface precipitation and virgae preceding it is not as clear as for the other sites. The enhanced sublimation at this site may delay the onset of surface precipitation. A position of the low-pressure system that causes precipitation over the MRR-PRO 06 and MRR-PRO 22 can, instead, be associated with virga over the MRR-PRO 23, at least until the environment reaches sufficient levels of saturation.

Overall this study demonstrates how the complex terrain south of PEA affects precipitation and its dependency on the local atmospheric conditions and the large-scale flow. The analysis conducted shows how a significant variability can be observed over a relatively small area, highlighting the importance of expanding the analysis of precipitation to locations beyond the boundaries of the scientific bases to understand the spatial variability of precipitation and accumulation over the ice sheet.





This type of analysis would be particularly beneficial in regions of complex topography, such as the Antarctic Peninsula or the Transantarctic mountain range.

While the current study examined the occurrence of virga and surface precipitation, the analysis could be extended to the accumulation at the three MRR-PRO sites. To do so, the first necessary step would be to further investigate the inter-calibration of the three MRR-PRO in order to reliably analyze and compare the intensity and amount of precipitation at the three sites.

Finally, not all the measurements collected during the PEA Precipitation Experiment (POPE) have been utilized in this study. Future studies could benefit from the polarimetric radar products provided by the scanning X-band Doppler weather
radar deployed close to the base, alongside the profiles collected by a W-band Doppler cloud radar near PEA, to characterize the microphysical processes governing precipitation formation and evolution.

*Data availability.*   The radar measurements included in the analysis are part of the dataset described by (Ferrone and Berne, 2023a) and are available for download at https://doi.org/10.5281/zenodo.7428690 (Ferrone and Berne, 2023b). The output of the atmospheric simulations can be requested by contacting the corresponding author.

**Appendix A: Precipitation in the austral summers before 2019/2020**

The results presented in the current manuscript are based on a dataset of approximately one month of radar measurements collected between 15 December 2019 and 17 January 2020. Due to the short time span, it is necessary to contextualize these observations by comparing them with atmospheric simulations and reanalysis covering a longer period.

First, we decided to compute the precipitation frequency and total accumulation over a period of 20 years, selecting the same
interval of dates as the one covered by the MRR-PRO measurements. To perform this analysis, we used the total precipitation available in the ERA5 reanalysis, extracted from the grid point closest to the location of PEA. The results are summarized in Figure A1.

According to the results shown in panel A1.b, the summer of 2019/2020 experienced the highest frequency of precipitation, occurring in almost half of the time steps in the period considered. This value is close to the one recorded during the previous
summer, but significantly higher than all the remaining entries in the time series. The total accumulation recorded during the summer of 2019/2020 instead reaches only the quantile 0.75 of the distribution of the two decades.

These results suggest that precipitation was significantly more frequent than usual during the period of deployment of the MRR-PRO transect. This frequency of occurrence, however, did not correspond to a similarly exceptional total accumulation. Higher amounts have been recorded during a few of the previous summers despite the less frequent precipitation, indicating
that fewer or shorter events, with relatively high intensity, may have contributed to the accumulation more significantly than during the summer of 2019/2020.

The second comparison presented in this section involves a set of three atmospheric simulations, performed using the WRF model, covering the austral summers of 2014/2015, 2015/2016, and 2016/2017. The model setup follows the one described in



section 2.2. The simulations were originally performed before the measurement campaigns, and have been used to decide the
location of the three MRR-PRO, as described in Ferrone and Berne (2023a). In particular, we used the spatial pattern of total
precipitation at the ground to select a few regions of interest for the transect. All simulations identify a local maximum in the
accumulation in the immediate proximity of the mountain complex between MRR-PRO 06 and 22, and a local minimum in the
valley where the MRR-PRO 23 is located.

Since sublimation plays a central role in the discussion presented in this manuscript, we decided to analyze the profiles
of $RH_i$ from the three model outputs during surface precipitation, virga, and clear-sky conditions. The profiles have been
computed following a procedure analogous to the one described in section 3.2, and the results are shown in Figure A2. With
only a few exceptions, it is possible to notice the presence of the relatively dry layer in the height levels below 3 km a.m.s.l.,
matching the one visible in the bottom row of panels of Figure 7.

The consistency with which this layer appears in the simulations indicates that its effect on the sublimation of falling
hydrometeors, and the resulting spatial distribution of precipitation, could be a recurring factor during the austral summer in
this region. Differences in the frequency of occurrence and intensity of precipitation among the years in the series may lead to
a different impact of this layer on snowfall events. Future measurement campaigns, involving weighing precipitation gauges or
radar measurements, may be able to characterize those differences.

**Appendix B:  Comparison between the precipitation occurrence in WRF and in the MRR-PRO measurements**

In section 3.2 we analyzed the profiles of atmospheric variables above the three sites extracted from the atmospheric simula-
tions. In this appendix, we compare the frequency of occurrence of virga and surface precipitation simulated by the model to
the ones recorded by the three MRR-PRO. To match the temporal resolution of the WRF output, only the radar measurements
averaged over a window of 1 hour have been used for the comparison. Additionally, to account for the limited sensitivity of the
MRR-PRO, only time steps in which WRF outputs a precipitation rate greater than 0.01 mmh$^{-1}$ (of liquid water equivalent)
have been kept in the surface precipitation category for the comparison. We used the formula proposed by Souverijns et al.
(2017) to convert the $Z_{ea}$ measurements of the three MRR-PRO into snowfall rate, and all the three profilers measured a mini-
mum rate close to the chosen threshold of 0.01 mmh$^{-1}$ (i.e. within an interval of 0.002 mmh$^{-1}$). The resulting frequency of
occurrence and ratios between virga and surface precipitation from the radars and from the model is shown in Figure A3. The
quantities displayed in each panel follow the same order as the ones in Figure 3.

The model overestimates the occurrence of both virga and surface precipitation, as seen in panels A3.b, A3.e and A3.f. The
increase in occurrence is particularly noticeable for virga at the MRR-PRO 06 and 22 sites, and for surface precipitation at the
MRR-PRO 23 site, as can be noticed in panels A3.c and A3.d. This overestimation results in the difference between the model
ratios and the radar derived ones in A3.a. However, the model preserves some of the differences between the three sites that
are relevant to the analysis presented in the manuscript. In particular, A3.e identifies the MRR-PRO 22 site as the one with the
lowest frequency of virga occurrence, and the the MRR-PRO 23 as the one with the lowest proportion of surface precipitation.



*Author contributions.* A.F. and A.B. designed the study. A.F. and A.B. deployed and maintained the instruments at PEA. A.F. processed the data collected during the measurement campaign. E.V. defined the setup for the WRF simulations. A.Z. performed the atmospheric simulations for the austral summer 2019/2020. A.F. prepared the manuscript with contributions from A.B., E.V., and A.Z., and supervision from A.B.

*Competing interests.* A.F., E.V., and A.Z declares that no competing interests are present. A.B. is associate editor for AMT.

*Disclaimer.* This research was funded by the Swiss National Science Foundation (grant number 200020-175700/1) the Swiss Polar Institute (Polar Access Fund 2019 and Exploratory Grant).

*Acknowledgements.* We would like to thank all the EPFL-LTE collaborators for their help in shaping the study and their contribution to the measurement campaign at Princess Elisabeth Antarctica. We are also grateful to the staff of the International Polar Foundation and all the
personnel of the base for their contribution to the deployment and maintenance of the scientific instruments on site.



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



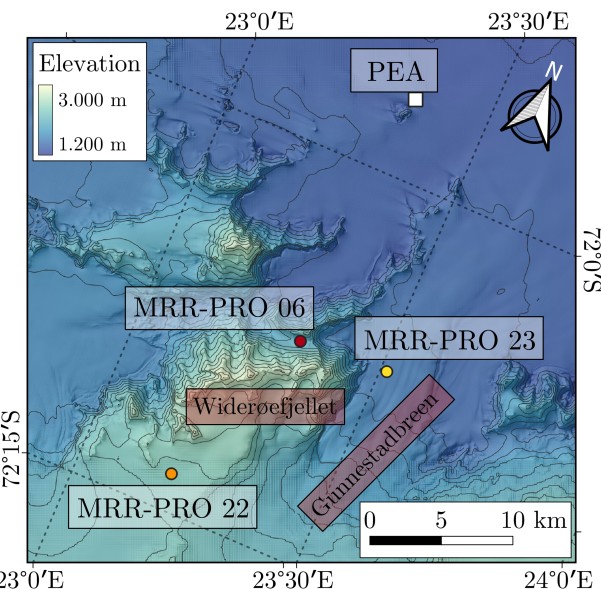

**Figure 1.** Elevation map (Howat et al., 2019) of the surroundings of PEA, including the section of the Sør Rondane Mountains directly south of the base. The white square shows the location of PEA, while the colored circles illustrate the locations of the three MRR-PRO: the MRR-PRO 06 in red, the MRR-PRO 22 in orange, and the MRR-PRO 23 in yellow. The mountain peak and glacier mentioned in the manuscript are denoted by text on a pale red background.



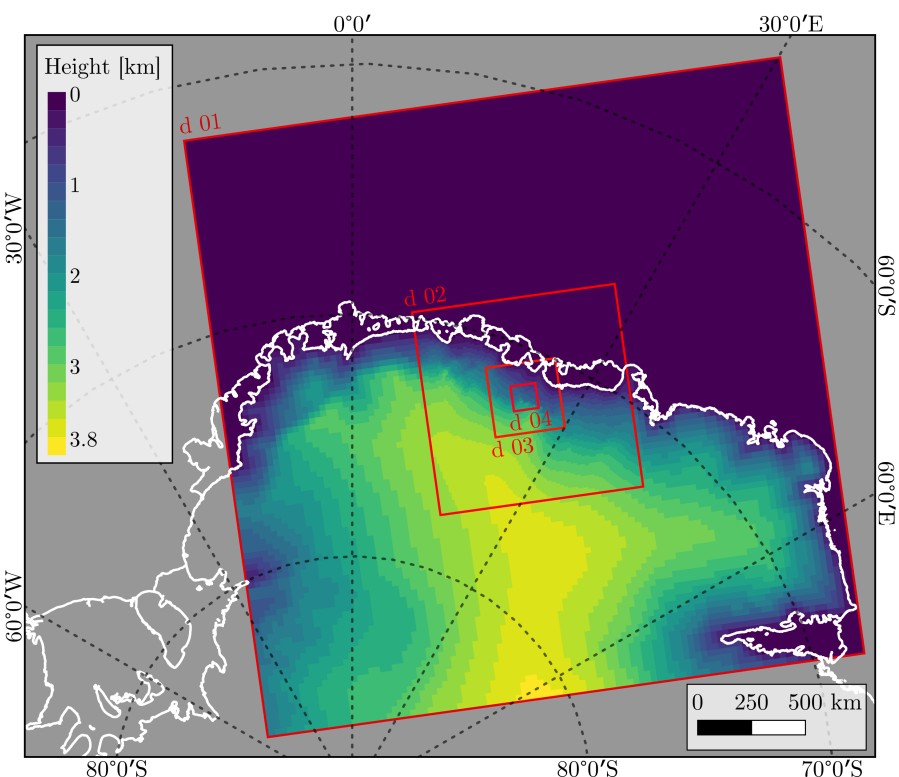

**Figure 2.** The extent of the four domains in the WRF simulations. The domains are numbered from the outermost (d01) to the innermost (d04). The color shading within each domain illustrates the terrain as seen by the model, with the appropriate horizontal resolution. The coastline and ice shelves' boundaries, from Gerrish et al. (2021), have been added for clarity purposes.

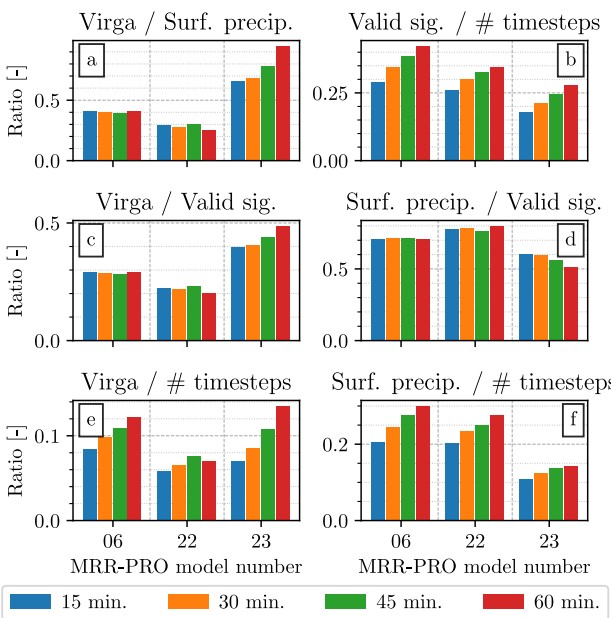

**Figure 3.** Summary of the results of the labeling of the averaged MRR-PRO measurements as virga or surface precipitation. Panel a shows the ratio between the number of time steps belonging to each category. Panel b displays the ratio between the number of time steps containing at least one valid measurement and the total number of time steps. Panel c (d) illustrates the ratio between the virga (surface precipitation) time steps and the number of time steps with a valid signal. Panel e and f show a similar set of ratios as panels c and d, using the total number of time steps at the denominator instead. The color of each bar denotes the duration of the averaging window. The total number of time steps used to compute the fractions varies in accordance with the window size: 3216 for the 15 minutes one, 1608 for the 30 minutes one, 1072 for the 45 minutes one, and 804 for the 60 minutes one.

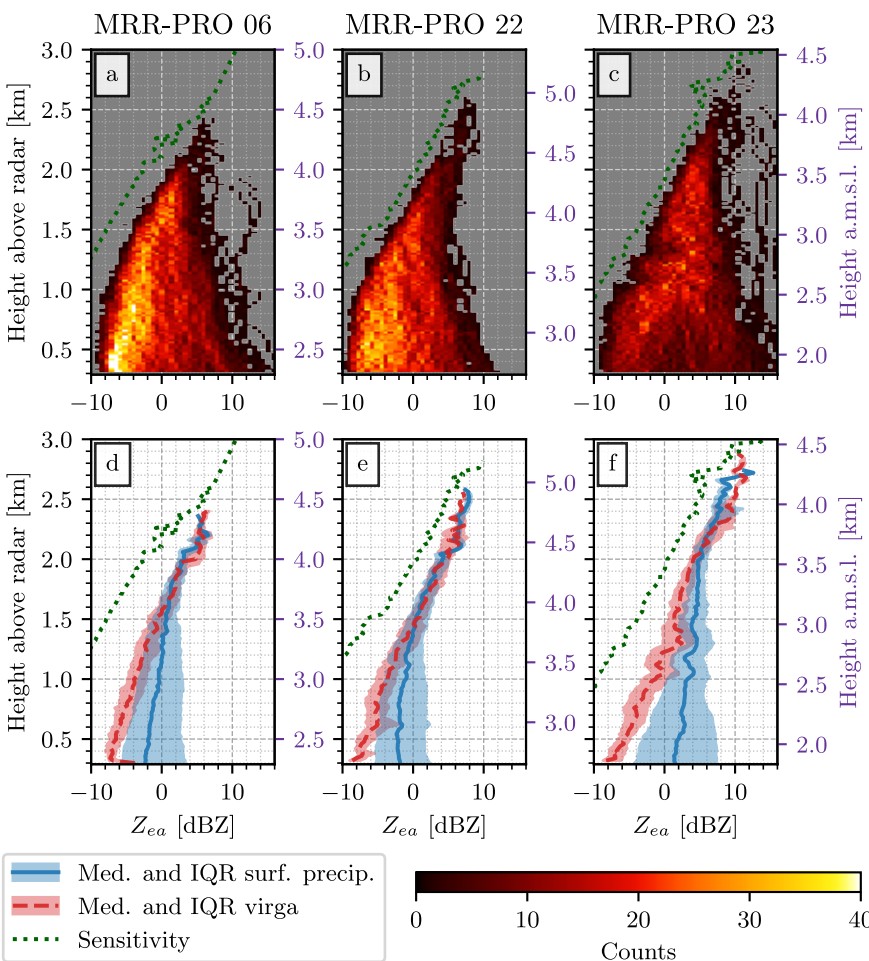

**Figure 4.** Averaged $Z_{ea}$ profiles from the three MRR-PRO in the transect, covering the period from 15 December 2019 to 17 January 2020. Panel a, b, and c show the 2-dimensional histogram of all averaged $Z_{ea}$ values versus height for each of the three MRR-PRO. The y-axis on the left of each panel displays the height above the first range gate, while the one on the right (in purple) shows the altitude above the mean sea level. The averaging windows chosen for the plot is 30 min. Panel d, e, and f illustrate the median and interquartile range of the averaged $Z_{ea}$ values during virga (in red) and surface precipitation (in blue) events. In all panels, the dotted green line denotes the sensitivity curve, computed using the quantile 0.01 of the empirical distribution.

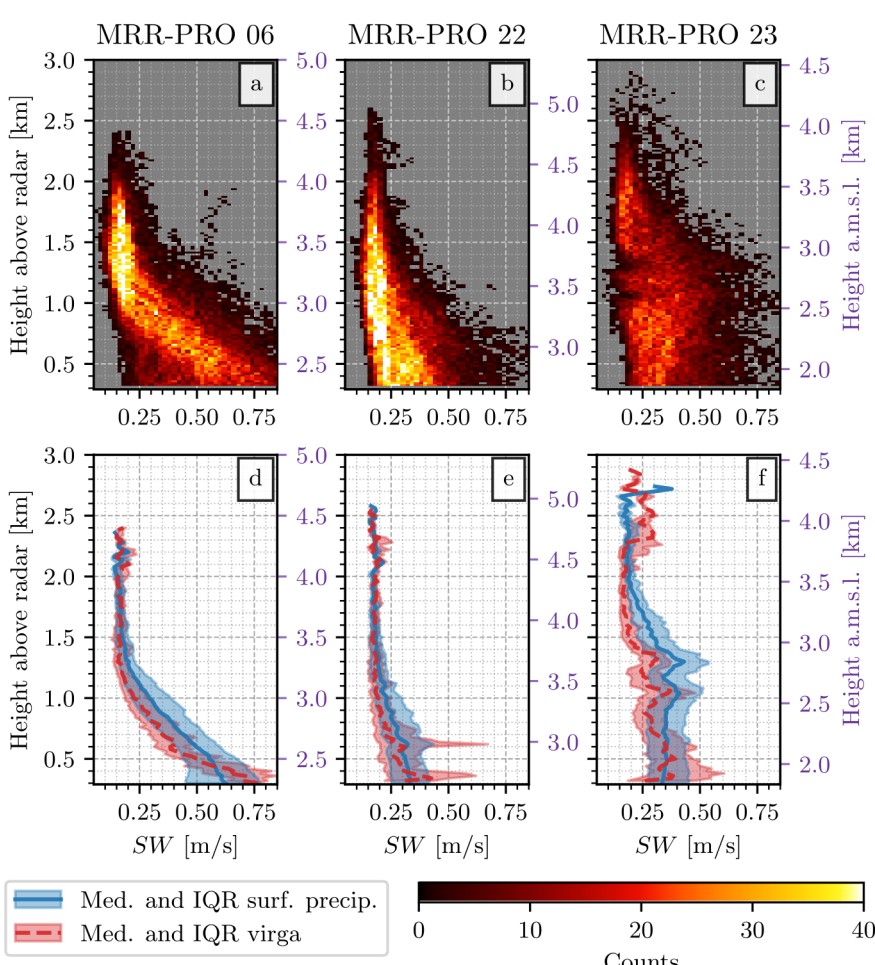

**Figure 5.** Averaged spectral width ($SW$) profiles. The figure is structured similarly to Figure 4.

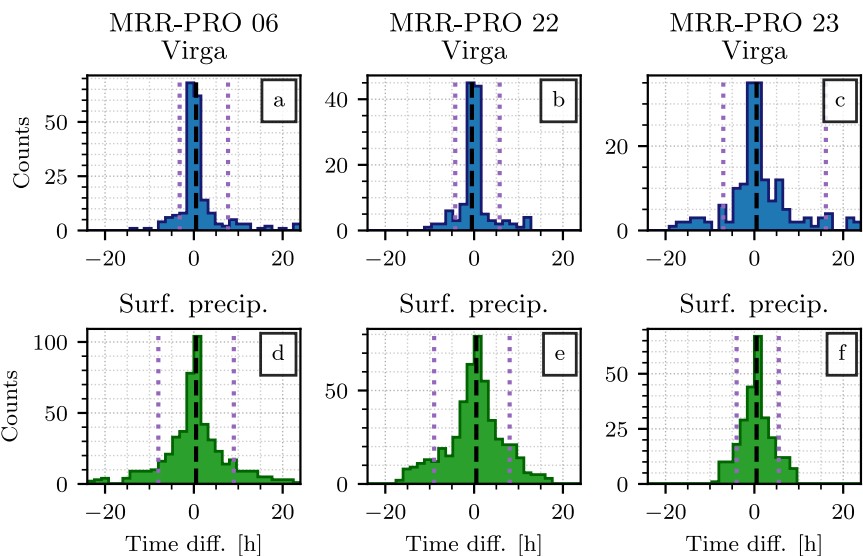

**Figure 6.** Histogram of the time lag between virga and surface precipitation time steps. In the first row (panels a, b, c), the time of each virga event is subtracted from the time of the closest surface precipitation event. The second row (panels d, e, f) is analogous to the first one, but with the roles of virga and surface precipitation inverted. Each column displays the statistics for a different site: MRR-PRO 06 in the first one (panels a, d), MRR-PRO 22 in the second one (panels b, e), and MRR-PRO 23 in the third one (panels c, f). The black dashed line denotes the median of each distribution, while the dotted purple lines indicate the quantiles 0.1 and 0.9.

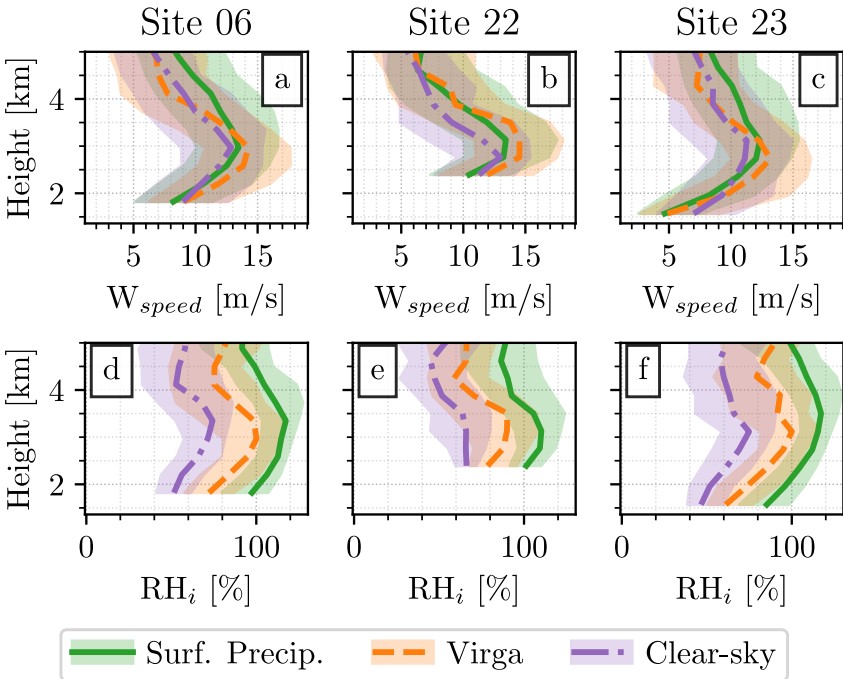

**Figure 7.** Median and IQR computed from the profiles of atmospheric variables extracted from the outputs of the WRF simulations at 1 km resolution. Each column of panels refers to a different site: MRR-PRO 06 (panels a, d), MRR-PRO 22 (panels b, e), and MRR-PRO 23 (panels c, f). Each row represents a different variable: horizontal wind speed in the top one (panels a, b, c) and relative humidity with respect to ice in the bottom one (panels d, e, f). The three different weather status are differenciated as follows: precipitation reaching the ground is denoted by the green continuous line, positive mixing ratio of solid hydrometeors without any accumulation at the ground (proxy for virga) by the dashed orange line, and clear-sky conditions by the dash-dotted purple line.



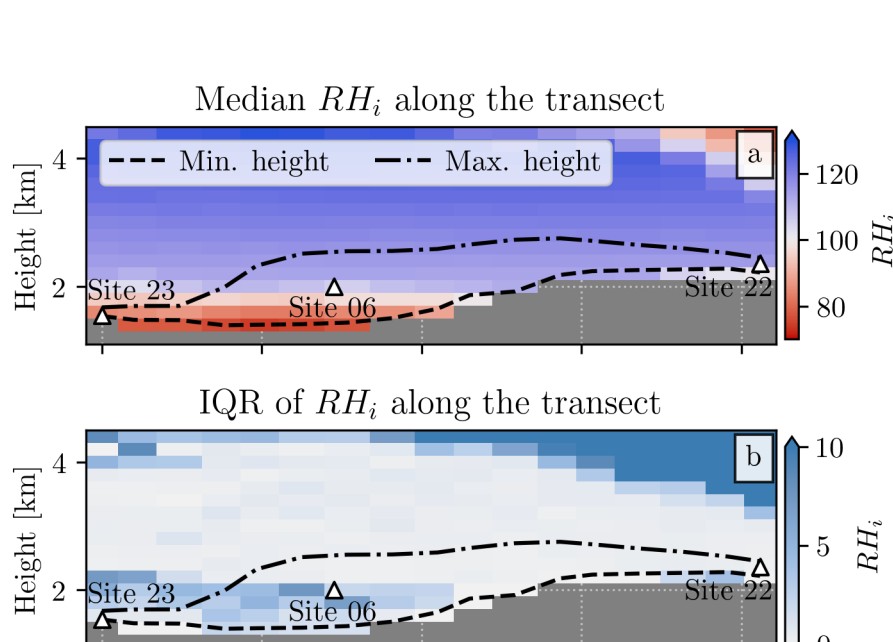

**Figure 8.** $RH_i$ values along the transect for the precipitation events on 06/01/2020 at 12:00 extracted from the outputs of the WRF simulations. The median (panel a) and interquartile range (panel b) have been computed on a 10 km padding of the line connecting the three MRR-PRO (i.e. 5 grid points on each side of the line). The dashed line represents the minimum of the terrain height along the transect, while the dash-dotted one shows the maximum.





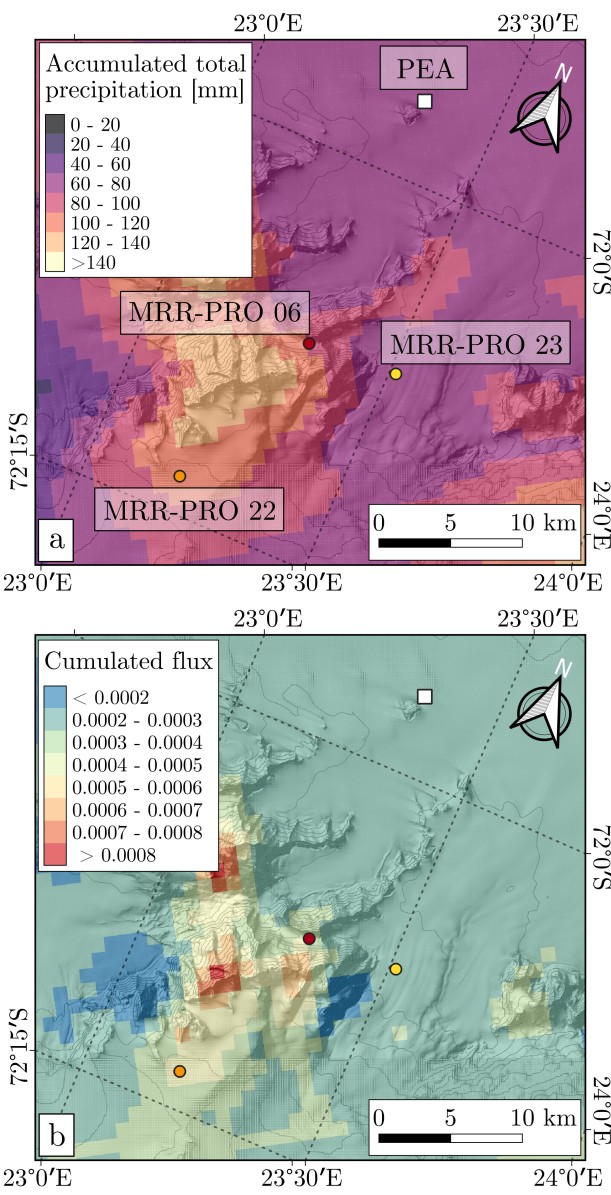

**Figure 9.** Toatal cumulated precipitation at the ground (panel a) and precipitation flux at 3500 m a.m.s.l. (panel b) from the WRF simulations. The location of PEA and the three MRR-PRO follows the same convention as Figure 1.



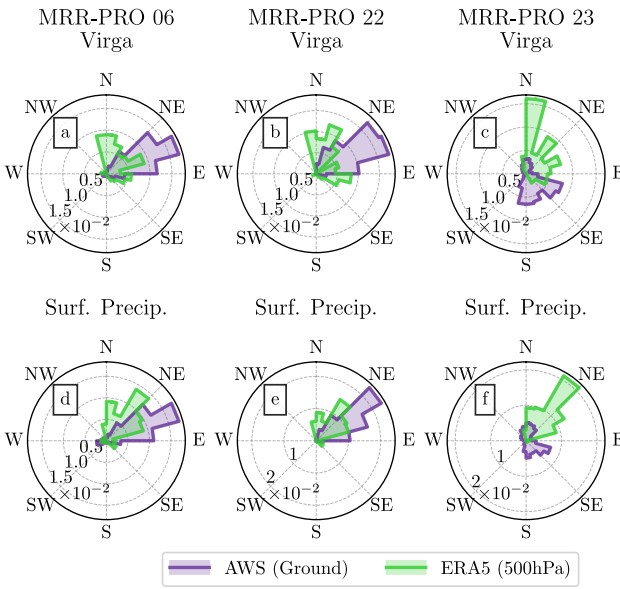

**Figure 10.** Frequency of wind direction at the three locations in the transect, divided in virga (panels a, b, c) and surface precipitation events (panels d, e, f), for the time period of deployment of the MRR-PRO. Each column of panels refers to a different site in the transect: MRR-PRO 06 in panel a and d, MRR-PRO 22 in panel b and e, MRR-PRO 23 in panel c and f. The measurements collected at the ground level by the AWS are displayed in purple, while the wind direction at 500 hPa from the ERA5 reanalysis are shown in light green.



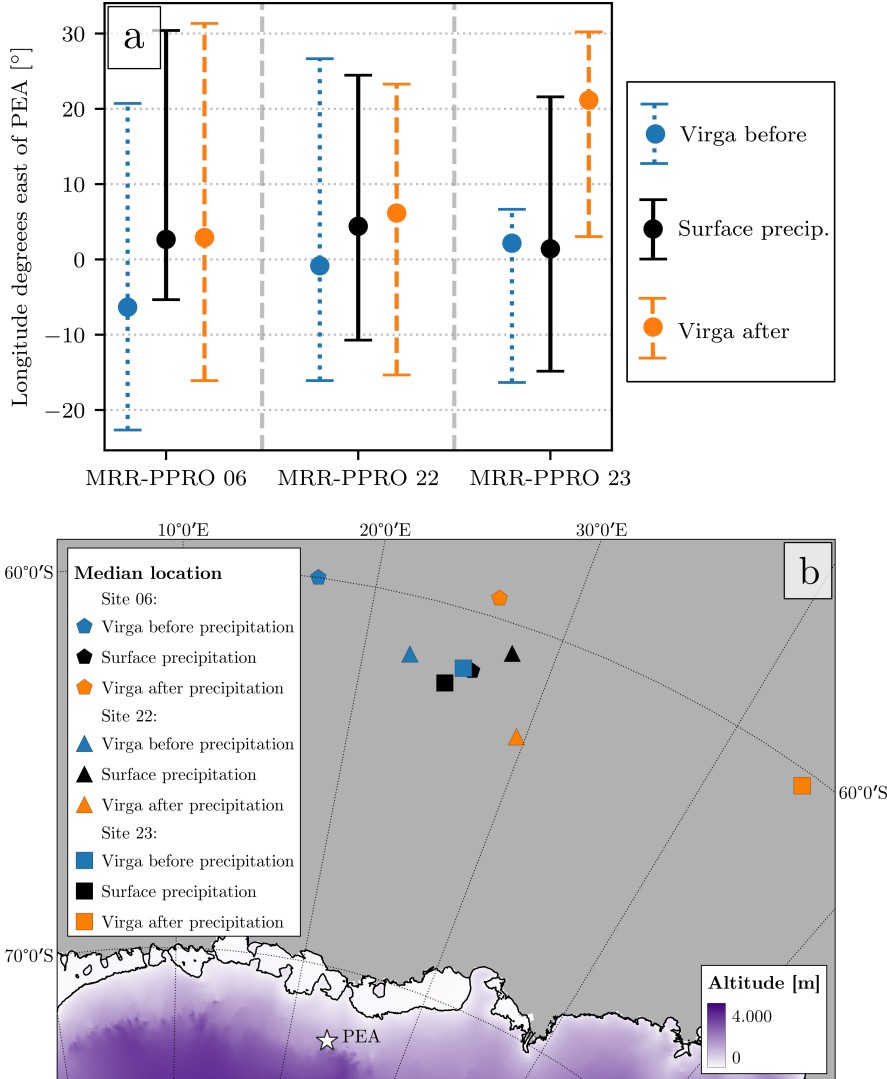

**Figure 11.** Location of the low pressure system along the Antarctic coastline during virga and surface precipitation events. Panel a shows the median (dots) and interquartile range (vertical lines with caps) of the longitude of the low pressure system relative to the longitude of PEA. Positive values on the y-axis represent location eastward of the base. At each of the three locations, the virga events before surface precipitation are denoted by the blue dotted line, the surface precipitation by the black solid line, and virga after the precipitation by the orange dashed line. Panel b shows the median latitude and longitude of the low pressure systems superimposed to a map of the region south of PEA. Pentagons indicate the locations associated to the MRR-PRO 06, triangles to the MRR-PRO 22, and squares to the MRR-PRO 23. The colors of each marker follows the same convention as panel a.

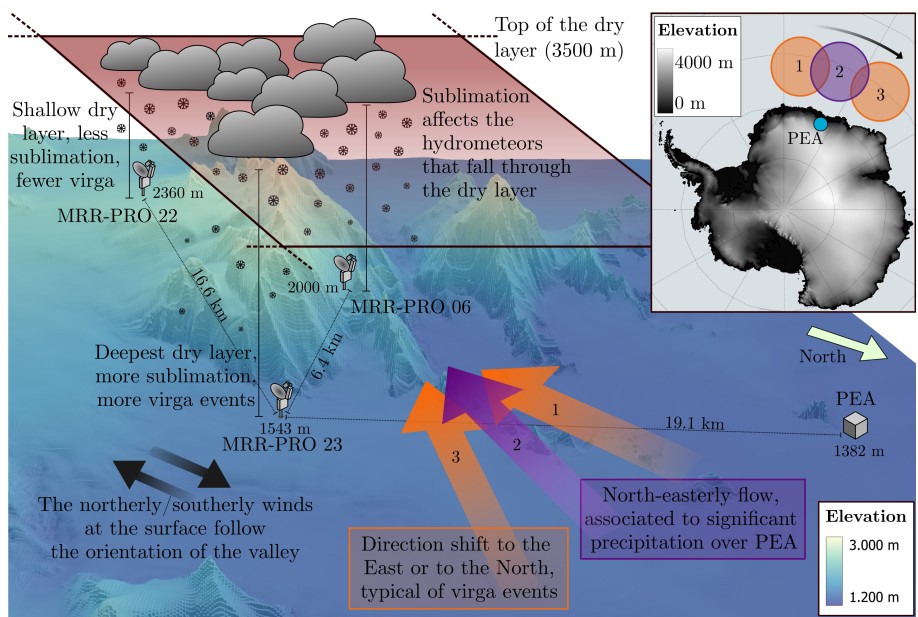

**Figure 12.** Conceptual model summarizing the mechanisms described in the study. The location of the three MRR-PRO (stylized radar symbols) and of the base (gray cube) are provided above a 3-dimensional representation of the terrain height (Howat et al., 2019). Dotted lines provide the approximate distance between the radars and the base. The semi-transparent red plane represent the top of the sub-saturated or dry layer, with continuous black lines denoting the different depth of the layer above the three sites in the transect. Stylized clouds and snowflakes highlight the different distance that falling hydrometeors have to travel through the dry layer to reach the ground. The location of low-pressure systems along the Antarctic coastline is shown as circles in the panel at the top right corner of the figure. The location associated to surface precipitation is shown in purple, while virga events are in orange. The numbers provide the order of the events: 1 for virga preceding the precipitation, 2 for surface precipitation, and 3 for virga following the precipitation. The wind direction at the base associated to the different cyclone locations is shown above the 3-dimensional terrain by arrows of the same color and numbers as the circles in the panel.





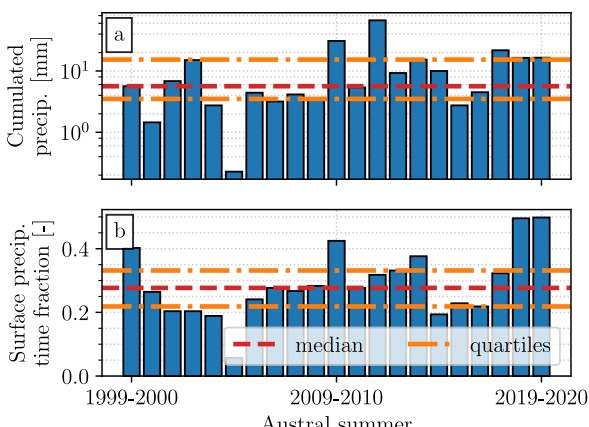

**Figure A1.** Total precipitation (panel a) and fraction of time steps in which precipitation is recorded (panel b) for the austral summers between 1999/2000 and 2019/2020. For each summer, only the time steps between 15 December at 00:00 UTC and 17 January at 12:00 UTC have been included in the analysis. The values recorded during each summer is represented by the blue bars, while the horizontal lines indicate the median (red dashed line) and interquartile range (orange dash-dotted line).

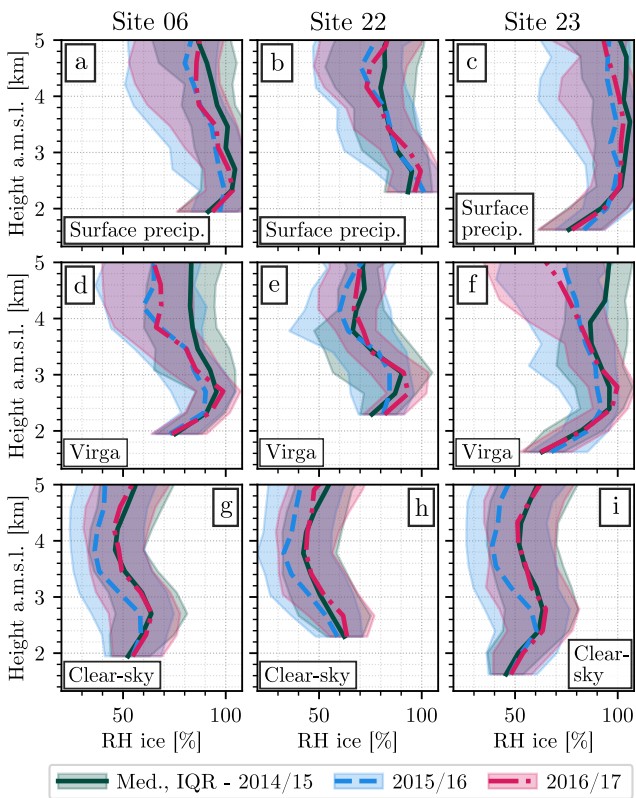

**Figure A2.** Profiles of $RH_i$ extracted from the outputs of the WRF simulations covering the Austral summers 2014/2015 (dark green, solid line), 2015/2016 (blue, dashed line), and 2016/2017 (red, dash-dotted line). The lines indicate the median value at each height, while the shaded area shows the interquartile range. Each column of panels refers to a different site: MRR-PRO 06 (panels a, d, g), MRR-PRO 22 (panels b, e, h), and MRR-PRO 23 (panels c, f, i). Each row represents a different weather status: surface precipitation (panels a, b, c), virga (panels d, e, f), clear-sky (panels g, h, i).





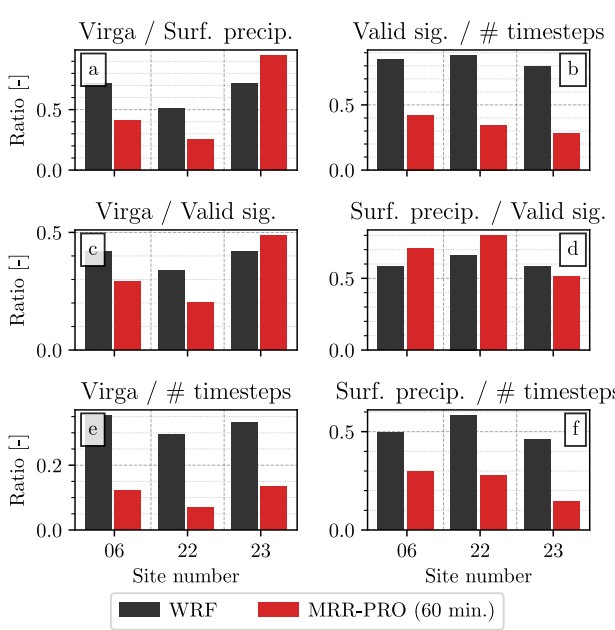

**Figure A3.** Comparison between the occurrence of virga and surface precipitation over the three MRR-PRO sites in the WRF simulations (dark gray bars) and in the radar measurements (red bars). The figure follows the same structure as Figure 3.