# Peer review of "Local spatial variability in the occurrence of summer precipitation in the Sør Rondane Mountains, Antarctica"

_EGUsphere, 2023_

## Author Comment (AC1)

**Reply to referee comments**

We thank the reviewers for the comments and suggestions, which helped us clarify part of the manuscript by including some missing information.

In the following, the comments from the reviewers are written in italics and highlighted in green

The line numbers mentioned in the replies are the ones in the track changes document, to which we will refer as "latexdiff" for brevity.

**1. Reply to RC 1**

*Line 218: you write: "0.25° × 0.25° horizontal grid". How many kilometers are these in Antarctica? Please add this information.*

Thank you for the suggestion, which makes the comparison between the ERA5 reanalysis and the other data sources used in the study more immediate.

The size of the ERA5 grid (9 km x 28 km) has been added in the manuscript (lines 236-237 of the "latexdiff").

*Line 316: Here you use a fixed delay in 15-minute increments. Why didn't you use autocorrelation to find the delay with the highest correlation on a better resolution?*

Thank you again for the suggestion. Unfortunately, while the proposed approach is valid in principle (and it would have allowed for a better distinction of the lags observed between the different sites), we do not have information on the occurrence of virga and surface precipitation at a better time resolution than 15 minutes. The identification of the precipitation type is performed after the temporal averaging (to avoid counting falling effects as virga) and the part of postprocessing following it. However, the 15-minute temporal step used in the current analysis is still sufficient, in our opinion, to highlight the difference in the timing of precipitation over the three sites in the transect.

*Line 331: Here you write: "at least 5 levels". What is the vertical resolution of the wrf model or how many levels does it have? Please mention somewhere (i.e. in the WRF section)*

We apologize for forgetting to include this important piece of information in the data section. The number of vertical levels (66) and information on the vertical discretization have been included in the revised manuscript (lines 208-211 of the "latexdiff").

For completeness, we include on the next page the plot of the median and IQR of the depth of the vertical levels at each model height below 5500m (the region on which most of our analysis focuses). This figure has not been included in the revised manuscript.

[Figure]

**Figure 1:** Depth of the model layers (x-axis) at the different layer heights (y-axis). The round markers show the median height, while the error bars indicate the interquartile range along both axes.

*Line 341: I think it would be beneficial if you would mention the comparison between MRR and WRF output earlier in this section.*

The sentence has been moved to the paragraph immediately below the list of weather types (lines 354-355 and 360-362 of the "latexdiff").

*Line 360: Please say a few words about the supersaturation (values above 100%). Are these real or typical values or is it model uncertainty?*

Thank you for pointing out this interesting feature of the $RH_i$ profiles.

At particularly low temperature (e.g. < -20°C) and under specific conditions, supersaturation may reach several tens of percent (Gierens et al, 2012, p. 148). For example, in Vignon et al., 2019, $RH_i$ values close to 110% have been observed in the profiles recorded by radiosondes at different Antarctic bases. Even though it is possible that WRF slightly overestimates $RH_i$ in our case, it is still likely that supersaturation can occur above PEA and the surrounding region.

In our opinion, these high values of $RH_i$ in some parts of the profile may be linked to the presence of mixed-phase clouds, during which saturation with respect to liquid water is almost reached. The high supersaturation would result from the fact that saturation vapor pressure with respect to liquid water is higher than the one with respect to ice.

The WRF simulations indicate the presence of liquid water in the clouds. Starting from the values of the "cloud water mixing ratio" from the WRF output, we selected the time steps in which the model simulates precipitation at the ground and extracted the profiles above the locations of the three MRR-PRO. We computed the median, quantile 0.75, and quantile 0.9 at each height and produced the figure shown on the next page.

[Figure]

**Figure 2:** Median (solid line with round markers), quantile 0.75 (darker area), and quantile 0.9 (lighter area) of the cloud water mixing ratio in the WRF simulations above the three MRR-PRO during precipitation events.

Even though in the majority of the profiles the clouds do not contain liquid water (median value equal to 0 kg kg$^{-1}$), the higher quantiles show positive mixing ratios. The maximum is often at 3 km a.m.s.l., close to the maximum of $RH_i$ shown in the manuscript.

Moreover, the multi-angle snowflake camera deployed over the rooftop of PEA during the same measurement campaign recorded pictures of falling graupel in some instances. An example can be found in Figure 7 of Ferrone et al. (2023). The presence of this hydrometeor type further confirms the presence of mixed-phase clouds above PEA during some events.

We decided not to include this additional figure in the manuscript, but additional information on the supersaturation values have been added in section 3.4 (lines 376-389 of the "latexdiff").

*Line 394: "MRR06" should be "MRR-PRO 06", right? Please change.*

Thank you for the correction, we have added it to the manuscript (line 428 of the "latexdiff").

*Line 417: Here, I suggest to summarize the section with one sentence. Why the wind direction analysis is important. Or what was expected?*

Thank you for the suggestion, the section was indeed missing some concluding remarks. The wind direction analysis is important because it suggests the existence of some differences in the circulation above the transect between virga and surface precipitation events. It supports the analysis performed on the ERA5 sea-level pressure data, providing some information on the local effects of the different locations of the low-pressure system along the Antarctic coast.
A sentence summarizing the importance of the analysis has been included in the revised manuscript at the end of the section (lines 452-456 of the "latexdiff").

**2. Reply to RC 2**

*L86: The main results of this analysis (as outlined in Appendix A with "These results suggest that precipitation was significantly more frequent than usual during the period of deployment of the MRR-PRO transect. This frequency of occurrence, however, did not correspond to a similarly exceptional total accumulation") could be recalled here in one brief sentence, as they put the peculiar conditions of your period of interest in a broader climatic perspective.*

Thank you for the suggestion, the addition of a summary sentence better connects the content of the appendix to the main text. We added it to the revised manuscript (lines 88-91 of the "latexdiff").

*Fig.2 you may want to specify that contours are shown every 200 m here.*

The information has been added to the caption of the figure in the revised manuscript.

*L180-187: Does uncertainty could be quantify using WRF simulations? For instance, looking at the distribution of the the ratio of modelled precipitation rates at the atmospheric level closest to 300 m and at the surface, could be a first attempt, from a modelling perspective, to quantify the uncertainty introduced when considering precipitation at 300 m as surface precipitation.*

Thank you for the suggestion, we could indeed use WRF for quantifying the difference in precipitation at 300 m and at the ground. For each grid point, we computed the cumulated precipitation flux (same procedure as in section 3.4) at these two levels, and then computed the difference between the two. This difference has been divided by the value at the ground, resulting in a relative difference, shown in the figure below.

[Figure]

**Figure 3:** Relative difference in the precipitation flux at 300m and at the ground over the transect.

While some considerable differences can be observed over some of the mountains, the relative difference between the flux at the ground and the one at 300 m over the 3 MRR-PROs is between 0.1 and 0.3 (10% and 30% if we express the relative difference as a percentage).

To better examine this difference, we looked at its temporal evolution at the three sites. In the figure below, we show the fluxes (top panel), their difference (middle panel), and the relative difference (bottom panel) for the MRR-PRO 06, from the start of the time series to the end of 2019.

[Figure]

**Figure 4:** Time series of precipitation fluxes and their difference above the MRR-PRO 06 from the start of the deployment period to the end of 2019. The top panel shows the precipitation flux at the ground (dashed orange line) and at 300m above the surface (dash-dotted blue line). The middle panel shows the difference between the two fluxes (300m minus ground). The bottom panel shows the relative difference (difference divided by the flux at the ground) as a percentage value.

We decided to show only 2019 in the figure to allow a better readability of the curves. The fluxes for the MRR-PRO 06 during 2020, and for the other radars during both 2019 and 2020 have been included later in this document (section 4).

The difference between the two levels varies between events and is particularly large for some of them. If the precipitation at 300m is used to estimate the accumulation at the ground, the difference in the final estimate may be significant, according to these results.

However, the impact on the detection of virga and surface precipitation should be more limited. The events in which the difference between the two fluxes is more marked are also the ones with the highest value of flux and, despite the difference, the lowest of the two values is often higher than the fluxes observed in other events. Therefore, the difference between the two fluxes should not result in many cases of mislabelling as virga or surface precipitation when using 300 m instead of the ground level in the analysis.

It should be noted that, while this result applies to the majority of the events, there are a few noticeable exceptions:

- the last event of the 2019 time series for the MRR-PRO 06;
- the last event at the end of the 2019 time series for the MRR-PRO 23 (shown in section 4 of this document);
- one event in the second part of the 2020 time series for the MRR-PRO 06 (shown in section 4 of this document).

While we deemed this analysis too long to be fully included in the revised manuscript, we included a few key results in section 2.1.1 (lines 105-115 of the "latexdiff").

*Section 2.2: How many levels are used to represent the atmosphere? How are they discretized along the atmospheric column? This must be added to Section 2.2 for exhaustivity.*

Thank you for noticing that the information was missing. We added the number of vertical levels (66) and some information on the discretization in the text (lines 208-211 of the "latexdiff").

For completeness, we made a figure that shows the depth of each level at different height, which we included in the reply to RC1 (2 pages before in the current document). The figure has not been included in the manuscript.

*L259: correct "on one hand"*

The sentence has been corrected (line 278 of the "latexdiff")

*L275-276: Is this analysis procedure also shared with Jullien et al. (2020)? If yes, you should mention it so the complementarity between all these different studies can be strengthen.*

While Jullien et al. (2020) perform a similar analysis in regard to the location of low-pressure systems and use measurements from the MRR-2, their analysis does not include statistics computed over time-averaged profiles of $Z_{ea}$. Therefore, we decided to keep the citation to their study only in sections 2.3 and 3.5.2 (where we use the sea-level pressure from ERA5 to investigate the location of low-pressure systems) and section 3.3 (which focuses on the time difference between virga and surface precipitation).

*L375: Is 1 m/s a typical sedimentation velocity for solid hydrometeors? Any reference to support your choice?*

Thank you for suggesting a clarification on this fallspeed value, for which we did not provide any source. In the revised manuscript, we included a short clarification (lines 405-408 of the "latexdiff"), citing the study by Durán-Alarcón et al. (2019), which observes similar vertical velocities from radar data at PEA. Since we assume a constant fallspeed across the whole region, changes to its value should not affect the spatial patterns that we discuss in section 3.4.

*Fig. 7: Maybe specify that the Height in the vertical axis is given in km above sea level*

The information has been added to the caption of the figure in the revised manuscript.

**3. Additional changes to the manuscript**

The following corrections have been included in the revised manuscript:

- Modified two citations to the previous scientific literature in the introduction (lines 43-44 and 46 of the "latexdiff");
- Corrected the meaning of the POPE acronym (line 563 of the "latexdiff");
- Removed the duplicated "https://doi.org" from the references.

**4. Additional figures**

In this section we include a series of figures to complement the reply to RC 2.

[Figure]

**Figure 5:** As Figure 4, but for the 2020 time series.

[Figure]

**Figure 6:** As Figure 4, but for the MRR-PRO 22.

[Figure]

**Figure 7:** As Figure 5, but for the MRR-PRO 22.

[Figure]

**Figure 8:** As Figure 6, but the MRR-PRO 23.

[Figure]

**Figure 9:** As Figure 7, but the MRR-PRO 23.

**5. References**

Durán-Alarcón, Claudio, Brice Boudevillain, Christophe Genthon, Jacopo Grazioli, Niels Souverijns, Nicole PM van Lipzig, Irina V. Gorodetskaya, and Alexis Berne. "The vertical structure of precipitation at two stations in East Antarctica derived from micro rain radars." *The Cryosphere* 13, no. 1 (2019): 247-264.

Ferrone, Alfonso, and Alexis Berne. "Radar and ground-level measurements of clouds and precipitation collected during the POPE 2020 campaign at Princess Elisabeth Antarctica." *Earth System Science Data* 15.3 (2023): 1115-1132.

Gierens, Klaus, Peter Spichtinger, and Ulrich Schumann. "Ice supersaturation." *Atmospheric Physics: Background–Methods–Trends*. Berlin, Heidelberg: Springer Berlin Heidelberg (2012): 135-150.

Jullien, Nicolas, Étienne Vignon, Michael Sprenger, Franziska Aemisegger, and Alexis Berne. "Synoptic conditions and atmospheric moisture pathways associated with virga and precipitation over coastal Adélie Land in Antarctica." *The Cryosphere* 14, no. 5 (2020): 1685-1702.

Vignon, Étienne, Olivier Traullé, and Alexis Berne. "On the fine vertical structure of the low troposphere over the coastal margins of East Antarctica." *Atmospheric Chemistry and Physics* 19.7 (2019): 4659-4683.